# Vascular Morbidity and Mortality in Craniopharyngioma Patients—A Scoping Review

**DOI:** 10.3390/cancers16061099

**Published:** 2024-03-08

**Authors:** Julia Beckhaus, Carsten Friedrich, Hermann L. Müller

**Affiliations:** 1Department of Pediatrics and Pediatric Hematology/Oncology, University Children’s Hospital, Carl von Ossietzky Universität Oldenburg, Klinikum Oldenburg AöR, 26133 Oldenburg, Germany; julia.beckhaus@uni-oldenburg.de (J.B.); carsten.friedrich@uni-oldenburg.de (C.F.); 2Division of Epidemiology and Biometry, Carl von Ossietzky Universität Oldenburg, 26129 Oldenburg, Germany

**Keywords:** craniopharyngioma, vascular disease, irradiation

## Abstract

**Simple Summary:**

Craniopharyngiomas are rare brain tumors that can occur in children and adults, who are usually treated with surgery and radiotherapy. These procedures and their resulting long-term consequences, such as morbid obesity, can increase the risk of vascular complications. This scoping review assesses the available evidence regarding vascular events associated with the tumor and its treatment. This review includes an overview of available studies and proposes avenues for further research. However, additional prospective observational studies are required to assess potential risk factors for cardio- and cerebrovascular events in craniopharyngioma patients.

**Abstract:**

Craniopharyngioma (CP) treatment, including surgery and radiotherapy, can have short- and long-term vascular side effects. Hypothalamic damage is related to morbid obesity and may increase the lifelong risk of experiencing vascular events in CP patients. This review summarized the available evidence regarding vascular complications in adamantinomatous or papillary CP patients, whatever their age at diagnosis. Three databases (Medline, CINAHL, Web of Science) were searched (06/2023) to retrieve eligible articles. The search was limited to peer-reviewed articles. Titles, abstracts, and full texts were screened by two independent reviewers, and data were extracted using a self-developed grid. Seventy-two studies were included in this review; the majority were case reports. Reported vascular sequela that occurred due to surgery were fusiform dilation of the carotid artery, stroke, vasospasm, hemorrhage, and aneurysm. Related conditions that emerged due to radiotherapy included Moyamoya syndrome and cavernoma. Cardiovascular morbidity and mortality often lead to hypothalamic obesity and metabolic syndrome in CP patients. Vascular damage is a rare complication of CP treatment. Surgical strategies should protect the surrounding hypothalamic and vascular structures. Patients receiving radiotherapy, particularly at a young age, should undergo magnetic resonance angiography monitoring to identify possible neurovascular sequela during post-treatment care.

## 1. Introduction

Located in the sellar and parasellar regions, craniopharyngiomas (CPs) describe rare, benign tumors (WHO grade 1), registering approximately 0.5 to 2 new cases per million people per year [1,2]. Among pediatric patients (patients under 18 years old), the adamantinomatous type (ACP) is the most common non-neuroepithelial intracranial tumor, representing approximately 5–11% of such tumors in this age group—the papillary type is more common in adults [3,4]. The age distribution has two peaks: one in children aged 5 to 15 years old and another in adults aged 50 to 70 years old [1].

Treatment often involves surgical resection using trans-sphenoidal or transcranial approaches, emphasizing hypothalamus-sparing surgical strategies [5]. Both the hypothalamus and the optic chiasm and critical neurovascular structures (f.e. arterial circle of Willis) are located in the surgical field. Vasospasm and further vascular complications are rare post-tumor resection complications [6]. After incomplete resection or if relapse or progress occurs, CP patients are often treated with radiotherapy [7]. High-precision photon therapy and proton beam therapy precisely target the (residual) tumor while protecting the surrounding structures and tissues [8]. Nonetheless, radiotherapy’s long-term adverse effects include vascular complications such as Moyamoya disease [9]. Moyamoya disease occurs in children and adult survivors of pediatric brain tumors and increases the risk of cerebrovascular events [10].

Due to their proximity to hypothalamic structures and the pituitary gland, both hypothalamic infiltration and surgical damage to the hypothalamic–pituitary axes (HPA) can cause hormonal and neuroendocrine dysfunction [11]. In patients with hypothalamic involvement and surgical hypothalamic lesions, BMI typically increases in the first year after surgery [12]. Neuroendocrine deficits, combined with hypothalamic damage, are covered by the umbrella term hypothalamic syndrome [13], which is an i.a. characterized by uncontrolled eating and morbid obesity [14]. Childhood obesity is related to adult cardiovascular disease (CVD) risk factors [15]. Thus, hypothalamus-involving CP patients have a lower 20-year overall rate of survival [16]. In general, adults who experienced childhood or young adult cancers have a 10-fold higher risk of experiencing stroke than their healthy siblings [17]. Severe obesity increases the risk of cardiovascular complications and may reduce life expectancy compared to those of healthy individuals of the same age [13,18,19]. Childhood-onset CP survivors must navigate decades of survival and need tailored treatment and multidisciplinary approaches to reduce the risk of late morbidity due to vascular sequela related to CP diagnosis and treatment [20].

This scoping review analyzes and summarizes the available evidence regarding vascular morbidity in CP patients at risk of experiencing problems related to the tumor, surgery, irradiation, or long-term sequela due to obesity or lifestyle factors. Due to the disease’s rareness and the resulting scarcity of data, a scoping review was the most appropriate method to review the existing studies and case reports. The review question sought to identify the quantity and types of vascular morbidity in adult and pediatric patients with childhood- and adult-onset CP in existing case reports and observational studies.

## 2. Materials and Methods

This scoping review was conducted in accordance with the JBI methodology for scoping reviews [21]. The review was not registered. This review was performed in accordance with the PRISMA (Preferred Reporting Items for Systematic Reviews and Meta-Analyses) guidelines.

### 2.1. Eligibility Criteria

The eligibility criteria for the record assessment process are described within the population, concept and context (PCC) framework (Table 1).

### 2.2. Types of Sources

This scoping review regarded case reports and observational studies as retrospective cohort studies and cross-sectional studies. No literature or narrative reviews were considered for inclusion.

### 2.3. Search Strategy

Our search strategy was devised to locate published and peer-reviewed studies. Initially, a restricted explorative search of Medline was performed to identify relevant articles on this topic. Terms extracted from the titles and abstracts of relevant articles, along with index terms characterizing the articles, were used to develop a full search strategy for Medline, CINAHL, and Web of Science (Appendix B). The search strategy, including all identified index terms and keywords, was adapted for each database. Additionally, the reference list outlining all included sources of evidence was screened to identify further studies. 

English- and German-language studies were included. Studies published before 1990 were excluded to reflect recent progress in imaging techniques, diagnostics, and treatment development. The searched databases were Medline, CINAHL, and Web of Science. All English- or German-language published up to 19 June 2023 were reviewed.

### 2.4. Study/Source of Evidence Selection

Following this search, all identified citations were collated and uploaded to EndNote version 20 (Clarivate Analytics, Philadelphia, PA, USA), and any duplicate studies were removed. Titles and abstracts were screened by two independent reviewers (JB and CF) for comparison with this study’s inclusion criteria. The full texts of the selected citations were compared to the inclusion criteria by two independent reviewers (JB and CF). Any disagreements between the reviewers at each stage of the selection process were resolved through the participation of an additional reviewer (HLM). The results of both the search and study inclusion process are shown in Figure 1.

### 2.5. Data Extraction

Data were extracted using a self-developed grid (Appendix A). Results on reference, title, study type, study size, childhood- or adult-onset CP, study period, type of vascular complication, attributed etiology, time since CP treatment (follow-up), outcome parameters, and results were extracted from the included studies. This step was undertaken by one reviewer (JB) and supervised by two additional reviewers (CF and HLM).

## 3. Results

### 3.1. Characteristics of Included Studies

Our systematic search retrieved 514 records. In total, 107 of 514 records were included in our study after abstract and title screening. Ten articles were excluded because they were published before 1990. Seventy-two articles were included in this review after full-text screening was performed. The majority (n = 58) were case reports or case series. Only 14 studies were observational studies.

### 3.2. Vascular Morbidity and Mortality Analyzed in Observational Studies

Though all 14 observational studies investigated cardiovascular complications, only 3 of 14 observational studies also assessed post-CP neurovascular complications. One study focused on cerebral infarction [22]. The studied populations, exposures, comparisons, and outcomes of the included studies are summarized in Table 2. Studies with reported estimates were classified as observational studies. Both childhood- and adult-onset CP patients were studied. One study comparing childhood- and adult-onset CP assessed the CVD risk, showing that adult-onset CP patients have a higher CVD risk than childhood-onset CP patients [23]. Multiple exposures were investigated: metabolic syndrome (MetS)/obesity [24,25], hypothalamic involvement (HI) [26,27], and growth hormone substitution therapy [28].

In their studies, Jung et al. demonstrated that MetS and HI led to cardiac autonomic dysfunction, increasing the risk of cardiovascular events in CP patients [24,27]. In a multicenter cohort study of childhood-onset CP patients, a higher degree of obesity was correlated with increased left ventricular wall thickness [25]. Holmer et al. found that not only was HI an important risk factor for CVD in CP patients, but female CP patients were also at a higher risk of experiencing a CVD event [26]. Compared to patients with a nonfunctioning pituitary adenoma, no statistically significant difference in the incidence hazard ratio (Table 2) exists for CP patients receiving growth hormone treatment, according to the analysis by Verweij et al. [28].

Pereira et al. assessed the prevalence of cerebrovascular accidents (CVA), myocardial infarction (MI), and transient ischemic attacks (TIA) in CP patients [29]. Cerebrovascular morbidity was common in CP patients (14% CVA, 2% TIA, 6% MI). The authors demonstrated that CVA was most common (40%) in female, premenopausal, and estrogen-deficient patients [29]. In the German KRANIOPHARYNGEOM 2007 study, an 11% prevalence of cerebral infarction was identified in patients with childhood-onset CP between 2007 and 2019 [22]. In Merchant et al.’s study, childhood-onset CP patients were treated with limited surgery and proton beam therapy (PBT), and CP patients treated with surgery and photon irradiation therapy (XRT) were compared at follow-up regarding the occurrence of vasculopathies [8]. Vasculopathies occurred in both groups—14 cases (7 in each group) were recognized at follow-up. Three-year cumulative incidences of 3.98% (SE 1.96) for XRT patients and 4.49% (SE 2.21) for PBT patients were reported based on retrospective analyses. The 5-year cumulative incidence was calculated to be 4.99% (SE 2.19) for XRT patients and 7.87% (SE 2.87) for PBT patients. In another cohort study, 2 of 37 (5.4%) childhood-onset CP patients had asymptomatic vasculopathy identified at follow-up [30]. These patients were treated with adjuvant radiotherapy. Using a machine learning approach, Qian et al. reported that 30 (24.2%) of 371 CP patients experienced venous thromboembolism (VTE). Age, craniotomy, CSF leakage, and a long surgical duration were risk factors for VTE.

Three studies compared the CP study population to the general population of the respective country and calculated the standardized cardiovascular/circulatory disease mortality (SMR) or incidence rate (SIR) [31,32,33]. All three studies reported a statistically significant increase in circulatory disease-specific SMR in the respective Swedish and Dutch populations (Table 2). Olsson et al. also reported an increased SIR for cerebral infarction (7.1 (95% CI 5–9.9)) in Swedish childhood-onset CP patients [32]. Based on the observational studies, extensive HI, obesity, and female sex were circulatory disease risk factors in CP patients. Compared to the general population, CP patients had increased circulatory disease-specific incidence and mortality.

**Table 2 cancers-16-01099-t002:** Summary of the included observational studies.

Patient Population	Number of Cases	Exposure	Comparison	Outcome Parameters	Result	Ref.
Childhood-onset CP	53	MetS	No MetS	Cardiac autonomic function and heart rate	Patients with MetS: lower levels of SDNN, TP, RMSSD, and HF	[24]
Childhood-onset CP	48	HI	No HI	Cardiac autonomic function and heart rate	Patients with extensive HI: lower levels of SDNN, TP, and HF. No interaction between HI and obesity for HRV	[27]
Childhood-onset CP	36	Obesity	No obesity	Cardiac status: TTE parameters	Degree of obesity in CP is correlated with an increased left ventricular wall thickness (r = 0.645, *p* < 0.001)	[25]
Adult-onset CP	24	Adult-onset CP	Childhood-onset CP	CVD risk	Adult-onset CP was associated with a higher Framingham risk score, atherosclerotic CVD 10-year score, and lifetime risk score than childhood-onset disease	[23]
Childhood-onset CP on long-term GHT	42	HI	No HI	CVD risk	HI was an important CVD risk factor in CP patients; there was a higher risk of female patients with CP having CVA	[26]
Adult-onset CP	291	GHT	Patients with nonfunctioning pituitary adenoma	CVA and CVD	Incidence hazard ratio: CVA: 0.99 (0.19–5.10); CVD: 1.17 (0.57–2.39); death: 1.1 (0.21–5.69)	[28]
Childhood- or adult-onset CP at Leiden Medical Center (the Netherlands)	54	CP	No CP	CVA, myocardial infarction, and TIA	Prevalence: 22% total CV morbidity, 14% CVA, 2% TIA, 6% MI; CVA was more prevalent in premenopausal estrogen-deficient women; mortality: 40% were CV complications	[29]
Childhood-onset CP	244	Patients after surgery for CP with cerebral infarction	Patients after CP surgery without cerebral infarction	Cerebral infarction	Prevalence: 11% of CP patients were identified as having cerebral infarction (2007–2019)	[22]
Childhood-onset CP	94	CP patients with limited surgery and PBT	Patients with usual care (surgery and XRT)	Vasculopathy	7 in PBT and 7 in XRT with vasculopathy during follow-up; 3-year cumulative incidences of 3.98% (SE 1.96) [XRT] and 4.49% (SE 2.21) [PTB]; 5-year cumulative incidences of 4.99% (SE 2.19) [XRT] and 7.87% (SE 2.87) [PTB]	[8]
Childhood- and young adult-onset CP	37	Patients with upfront adjuvant XRT	Patients with incomplete surgical resection	Vasculopathy	2 (5.4%) children in the adjuvant XRT group developed asymptomatic radiation-related vasculopathies at follow-up	[30]
Adult-onset CP	371	CP	Other tumor entities	Venous thrombo-embolism	30 (24.2%) of 371 CP patients had VTE; patients with increased age, specific tumor pathology (CP and chordoma), craniotomy, CSF leakage, and a long surgical duration were at a higher risk of developing VTE	[34]
Childhood- or adult-onset CP	224	CP	General population in the Netherlands	Circulatory diseases	SMR circulatory diseases 2.3 (95% CI: 1.1–4.5) mainly due to CVD	[33]
Childhood- or adult-onset CP	307	CP	General population in Sweden	Circulatory diseases, ischemic heart disease, CVD	SMR circulatory diseases 3.6 (95% CI 2.1–5.7); SMR ischemic heart disease 3.6 (95% CI 1.6–6.8); SMR cerebrovascular 5.1 (95% CI 1.5–12); SIR cerebral infarction 7.1 (95% CI 5.0–9.9)	[32]
Childhood- or adult-onset CP at Lund University Hospital (Sweden)	60	CP	General population in Sweden	Cardiovascular and cerebrovascular mortality	SMR 3.21 (95% CI, 1.29–6.61)	[31]

Abbreviations: CVA: cerebrovascular accident; SDNN: standard deviation of all normal R-R intervals; TP: total power; RMSSD: root mean square of the difference of successive R-R intervals; HF: high frequency; HRV: heart rate variability; TIA: transient ischemic attack; SMR: standardized mortality rate; SIR: standardized incidence rate; GHT: growth hormone treatment; TTE: transthoracic echocardiography; HI: hypothalamic involvement; CI: confidence interval; PBT: proton beam therapy; CSF: cerebrospinal fluid; Ref: reference.

### 3.3. Vascular Morbidity and Mortality Analyzed in Case Reports

#### 3.3.1. Neurovascular Sequela

Figure 2 summarizes the frequency of reported CP cases with neurovascular complications in the literature with regard to different associations. The fusiform dilatation of the carotid artery (FDCA) (n = 44) was the most frequently reported neurovascular complication, followed by vasospasm (n = 34), stroke (n = 30), Moyamoya disease (n = 26), aneurysm (n = 10), hemorrhage (n = 7), TIA (n = 6), and cavernoma (n = 5). Other rarer complications, with reported numbers under two cases, are not displayed. Associations either were surgery, irradiation, and the tumor itself or unspecified (Figure 2).

Figure 3 summarizes the short-, medium-, and long-term vascular complications of CP treatment via surgery and radiotherapy. Shortly after surgery (<1-year post-treatment), acute ischemic events and hemorrhage can occur. After radiotherapy, vasospasm and acute ischemic events were reported to be short-term complications (Figure 3). FDCA and aneurysm can be considered medium- or long-term consequences, occurring between one- and five-years post-surgery (Figure 3). Moyamoya syndrome and cavernoma are long-term complications following radiotherapy (more than 5 years post-treatment), whereas stroke/TIA and aneurysms are medium- to long-term complications of radiotherapy (Figure 3).

#### 3.3.2. Fusiform Dilatations of the Carotid Artery (FDCA)

FDCA was the vascular complication most frequently reported (n = 44) in CP patients. FDCA describes an abnormal enlargement or widening of the carotid artery. FDCA can occur in the internal (ICA) and the external carotid arteries (ECA). In the included studies, both dilations (ICA and ECA) were described. It was apparent that FDCA mainly occurred in childhood-onset CP patients (Table 3). In the German childhood craniopharyngioma registry, a prevalence rate of 2.4% (14 of 583 CP patients) was calculated for FDCA between 2001 and 2015 [35]. The time interval between occurrence and treatment varied between 4 months and 12 years after treatment (Table 3). FDCA was attributed to surgery in 95% of patients (42 out of 44 cases) and irradiation in 2.3% of patients (1 case) (Figure 2). FDCA was mainly reported after the use of transcranial surgery (Appendix A). Most patients did not present any clinical symptoms related to FDCA—headache was the initial symptom at FDCA diagnosis in three patients [36,37,38]. Due to carotid artery dilation, the disrupted blood flow could promote blood clots or emboli formation. Therefore, FDCA patients have an increased risk of stroke. However, for the cases reported in the literature, no ruptures or FDCA-related morbidity were described at follow-up.

#### 3.3.3. Vasospasm

Vasospasm was the second most frequently reported vascular complication (34 cases) in the included studies (Table 3). Cerebral vasospasms were observed during or after surgical resection (transcranial or trans-sphenoidal approach) of CP (Figure 2) in both pediatric and adult patients (Table 3). In the included case studies, vasospasm was primarily described as a side effect of radiotherapy (26 of 34 cases) (Figure 2). Vasospasms were diagnosed at a time interval of between 5 days and 2 weeks after irradiation (Table 3).

#### 3.3.4. Stroke/Transient Ischemic Attacks (TIA)

Overall, in 14 studies, 30 stroke cases and 6 TIA cases were reported in both childhood- and adult-onset CP patients (Table 3). Radiotherapy increases the risk of stroke and TIA (Figure 2). Stroke also occurred after surgery in six cases (Figure 2), following a transcranial approach (Appendix A). The time interval between stroke and CP diagnosis/surgery varied between 10 days and 20 years post-CP diagnosis (Table 3). Accordingly, stroke should be regarded as both a short- and long-term complication of CP treatment.

#### 3.3.5. Moyamoya Syndrome

Moyamoya syndrome was observed in 26 cases reported in 17 studies (Table 3). It mainly occurred in childhood-onset CP survivors. Moyamoya syndrome is a rare complication of irradiation treatment, affecting small blood vessels in the irradiated field. The arteries’ stenosis in Moyamoya syndrome has complications such as recurrent TIAs, stroke, seizures, and cognitive or developmental issues, though specific complications are dependent on the severity and location of the affected blood vessels. The occurrence interval for Moyamoya syndrome varied from 19 months to 30 years post-radiotherapy (Table 3). In most of the reported cases, information on long-term follow-up after Moyamoya disease diagnosis was lacking. Based on the included studies, no difference between photon and proton therapy was observed regarding the reported Moyamoya syndrome cases (Appendix A).

#### 3.3.6. Aneurysm

Unlike FDCA, aneurysms can involve more arterial vessels than just the carotid arteries. Aneurysms’ neuroradiological characteristics include arterial dilatations that do not have a fusiform shape, including saccular-shaped and “berry-like” arterial dilatations. Aneurysms were observed in 10 cases reported in six studies (Table 3). Aneurysms occurred in both childhood- and adult-onset CP patients. In adult-onset CP patients, aneurysms were side effects of surgery. In childhood-onset CP patients, aneurysms were diagnosed during an interval lasting between 8 months and 5 years after both surgical intervention and irradiation to treat CP (Table 3).

#### 3.3.7. Hemorrhage

Hemorrhage was mainly reported in adult-onset CP patients, with only one case reported in childhood-onset CP patients (Table 3). Overall, only seven cases reported in six studies were observed. The reported hemorrhages were exclusively observed as immediate and intra- and perioperative surgical complications (Figure 2) [39,40]. Two studies reported hemorrhage cases after using a transcranial approach, while one study reported such a case after using a trans-sphenoidal approach (Appendix A).

#### 3.3.8. Cavernoma

In patients with non-CP brain tumors, radiotherapy is a known risk factor for cavernoma development [41]. Cavernoma was reported in five cases between 3 and 10 years after radio-oncological treatment of childhood-onset CP occurred (Table 3). The clinical courses of the cavernomas were benign—none of the reported patients developed a symptomatic hemorrhage [42,43]. However, children aged under 10 years old, when receiving irradiation, have an increased risk of developing cavernomas [42]. Although patients with cavernomas typically do not present clinical symptoms, hemorrhage is a serious complication. Magnetic resonance imaging (MRI), including magnetic resonance angiography (MRA), should be used consistently for extended follow-ups, especially in CP patients treated during early childhood [42].

#### 3.3.9. Other Neurovascular Sequela

Other neurovascular complications included hematoma (two cases post-surgery), radiation-induced large vessel cerebral vasculopathy such as arterial stenosis and occlusion (two cases post-radiotherapy), pseudoaneurysm (one case after subtotal resection and radiotherapy), cerebroartherosclerosis (one case post-radiotherapy), and intracranial venous thrombosis (one case post-surgery for recurrence) (Table 3). Lucas et al.’s study reported 9 post-surgery stenosis cases and 22 stenosis cases post-PBT for childhood CP—3 patients presented with dilated perivascular space years after PBT [44]. Only hematoma was reported in both childhood- and adult-onset CP patients. All other complications only occurred in childhood-onset CP patients (Table 3).

**Table 3 cancers-16-01099-t003:** Reported cases with neurovascular complications.

Neuro- Vascular Complication	Number of Cases	Number of Reports	Time Interval since CP Treatment [Min–Max]	Childhood-/Adult-Onset	References
FDCA	44	13	4 months–12 years	Mainly child-hood-onset (two adult-onset)	[35,36,37,38,45,46,47,48,49,50,51,52,53]
Vasospasm	34	3	5 days–2 weeks after XRT	Both	[54,55,56]
Moyamoya syndrome	26	17	14.5 months–30 years	Mainly childhood-onset; two cases of adult-onset CP	[36,43,57,58,59,60,61,62,63,64,65,66,67,68,69,70]
Stroke/TIA	30 strokes 6 TIA	14	10 days after surgery–20 years post-CP diagnosis	Both	[44,63,65,66,70,71,72,73,74,75,76,77,78,79]
Hemorrhage	7	6	At surgery	Mainly adult-onset; one case of childhood-onset	[39,40,66,80,81,82]
Aneurysm	10	6	At surgery (adult); 8 months–5 years (children) 2 cases post-XRT (children)	Both	[43,44,83,84,85,86]
Cavernoma	5	3	3 years–10 years	Childhood-onset	[42,43,44]
Other	stenosis (31) dilated perivascular space (3) hematomas (2) RLVCV (2) pseudoaneurysm (1) cerebroartherosclerosis (1) intra-cranial venous thrombosis (1)	4	9 stenosis post-surgery cases and 22 post-PBT cases 3 dilated perivascular space cases post-PBT 5–63 months post-surgery (hematoma) 3.8 and 1-year post-XRT (RLVCV) 5 months (pseudoaneurysm) 26 years post-XRT (cerebroartherosclerosis) Immediately after surgery for recurrence (intracranial venous thrombosis)	Childhood-onset (hematoma: both)	[44,87,88,89,90]

Abbreviations: FDCA: fusiform dilatations of the carotid artery; RLVCV: radiation-induced large vessel cerebral vasculopathy; TIA: transient ischemic attack; XRT: irradiation; PBT: proton beam therapy; CP: craniopharyngioma.

### 3.4. Cardiovascular Sequela (Case Reports)

#### 3.4.1. Cardiac Arrest

Probst et al. reported cardiac arrests observed during CP surgery in one childhood-onset CP case and one adult-onset CP case [91]. The two patients, 8 and 21 years old, were successfully reanimated during surgery after cardiac arrest.

#### 3.4.2. Deep Venous Thrombosis and Pulmonary Embolism

Deep venous thrombosis (DVT) was observed in four cases reported in two studies [92,93]. The patients had childhood-onset CP and were diagnosed with DVT two weeks, two months, and 13 years after surgical CP treatment. The DVT was localized iliofemoral (two cases) and located in the lower extremity with extension into the inferior vena cava (one case). According to the authors, different pathophysiological pathways occurred in each patient: oral contraception and family history (16-year-old patient), post-surgical complication (13-year-old patient), and hypernatremic dehydration (5-year-old patient). One patient (10-year-old boy) experienced pulmonary embolism after DVT, occurring 25 days post-surgery, and veno-arterial extracorporeal membrane oxygenation (VA-ECMO) was required [93]. Another patient (4-year-old boy) experienced a bilateral pulmonary embolism on day 17 post-surgery.

## 4. Discussion

Vascular complications are rare events in CP patients. However, long-term circulatory disease-specific mortality and morbidity are higher in these patients compared to the general population. In the case reports identified for inclusion in this scoping review, FDCA and vasospasm were the most frequently observed neurovascular complications post-CP treatment. The included observational studies found abnormal echocardiographic values in CP patients associated with the degree of obesity [24,25]. CP survivors have higher circulatory disease-specific mortality and incidence compared to the studied general populations in several European countries [31,32,33]. The main risk factors for vascular morbidity and mortality observed in observational studies are extensive HI, female sex, and obesity [26]. Morbid obesity is a frequent sequela in patients with childhood-onset craniopharyngioma [12]. Patients with anterior and posterior HI and HL are at the highest risk of developing obesity [94]. Morbid obesity can result in MetS, a well-known risk factor for CVD. Long-term aftercare for this high-risk group of CP patients is needed for early diagnosis and the prevention of possible CVD events. Carotid artery, middle and anterior cerebral artery, posterior connecting artery, posterior cerebral artery, and basilar apex injuries were reported as intra- and perioperative vascular complications in CP. Due to these tumors’ tendency to encase in or adhere to both larger and smaller vessels, careful identification and microdissection of the tumor away from vessels is a key challenge of CP surgery. FDCA can occur immediately after surgery due to carotid manipulation or arterial wall weakening. Furthermore, aneurysm and hemorrhage can occur during surgery. Besides sparing the hypothalamic structures, surgeons should protect surrounding vessels to prevent vascular injuries. The best craniopharyngioma treatment approach typically includes a combination of strategies such as surgery, radiotherapy, and close monitoring. Furthermore, individual vascular complication risk factors increase the likelihood of an event. Consequently, vasculopathy development is multifactorial. Vasospasm, Moyamoya syndrome, and cavernoma were described as complications associated with radiotherapy. Vasospasm occurs shortly after radiotherapy (up to two weeks); Moyamoya syndrome can be a long-term complication of using radiotherapy to treat CP. One patient was diagnosed with Moyamoya syndrome 30 years after receiving CP treatment [59]. Usually, Moyamoya syndrome has no clinical symptoms. During follow-up, especially after radiotherapy, clinicians should be aware of this rare sequela and inform patients of their increased risk of experiencing neurovascular events due to Moyamoya syndrome.

Cavernomas can occur after radiotherapy treatment for brain tumors. Two childhood-onset CP cases with cavernoma were reported in the literature [42,43]. In a retrospective cohort of childhood cancer survivors treated with cranial radiotherapy, a minority of cavernomas (3 of 36 (8%) included patients) were classified as high risk for hemorrhage [41]. No symptomatic hemorrhage was reported in this cohort, and most individuals (92%) were classified as being at low risk of developing cavernoma [41]. However, more cases of radiotherapy-induced cavernoma with longer follow-ups are required to draw risk-related conclusions.

The implementation of MRA during and after CP treatment must be discussed. While FDCA, stroke/TIA, and aneurysms occur shortly after surgery, the radiation-related vasculopathies manifest as a late sequela. Lucas et al. reported no added radiotherapy-related risk of vascular morbidity in their patients [44]. However, further studies using screening for radiation-related vasculopathies are needed to evaluate the diagnostic value of routine MRA in aftercare. Furthermore, the clinical relevance of MRA-confirmed vasculopathies needs to be examined since many patients had abnormal angiographies but no symptoms or further complications.

Compared to previously published reviews, this scoping review identified and systematically summarized the available evidence regarding the quantity and different types of vascular damage in adult and pediatric patients with childhood- and adult-onset CP in existing case reports and observational studies. Jamshidi et al. reviewed the neurosurgical literature for FDCA after pediatric craniopharyngioma [95]. The authors also postulated that FDCA is a rare complication of CP surgery. If FDCA occurred, the course was innocuous, and no ruptures were described in the literature [95]. However, FDCA was the most frequently described complication among the studies included in our scoping review. Erfurth also reported the increased prevalence of hypertension, other cardiovascular morbidities, and MetS in CP patients in a narrative review [13]. CP patients more often receive treatment for CVD, as well as anti-hypertensive, anti-diabetes treatment, and lipid-lowering drugs. Steinbok reviewed the literature on cerebrovascular abnormalities in CP patients [9]. Linking cerebrovascular problems to surgery or radiotherapy, the author described the possibility of carotid pseudoaneurysm occurring as a result of surgery to remove a tumor adherent to the carotid artery. Furthermore, the author reported on internal carotid artery stenosis, middle cerebral artery stenosis, anterior cerebral artery stenosis, Moyamoya syndrome, stroke, and cavernoma caused by radiotherapy, consistent with our findings.

This scoping review has certain strengths and limitations. To the best of our knowledge, this is the first scoping review of the available evidence for vascular complications in childhood- and adult-onset craniopharyngioma patients. We used a broad systematic search of three databases and independently assessed the eligibility of studies by employing two reviewers. However, for language, our review was limited to articles published in German or English. Although this scope covers most scientific articles, we cannot rule out missing relevant information published in other languages. The data were extracted by one reviewer, potentially increasing the chance of missing details. Due to the disease’s rareness, the included evidence is limited to case reports, small sample sizes, and heterogeneous study groups.

## 5. Conclusions

In conclusion, vascular damage mostly occurs post-surgery or as a late side effect of radiotherapy in CP patients. Stroke, aneurysms, FDCA, and Moyamoya syndrome were the complications most commonly reported in the literature. While this review does not establish causality, it suggests a plausible association between vascular lesions and CP treatment, which should be taken into consideration in clinical practice. Prospective cohort studies are needed to assess the incidence of vascular sequela in this population. Furthermore, risk factors before and after CP treatment leading to vascular morbidity should be assessed in future studies. Target emulation trials are needed to compare the risk of vascular complications after different surgical approaches to treating CP. Follow-up care programs must consider the lifelong risk of CVD in CP patients. Tertiary prevention programs are required for CP survivors to reduce the risk of obesity-related cardiovascular events. In patients prone to neuro-cerebrovascular complications, especially after radiotherapy treatment, MRA should be applied at follow-up. Children who receive radiotherapy for CP early in life should be monitored regularly since they will experience decades of survival.

## Figures and Tables

**Figure 1 cancers-16-01099-f001:**
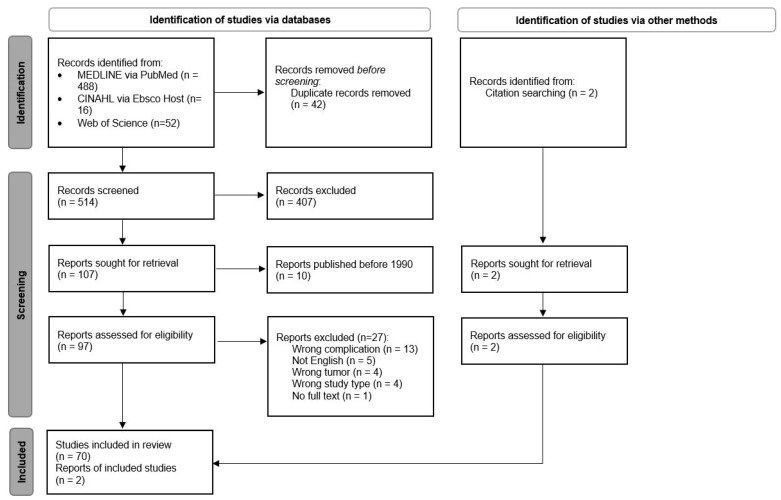
PRISMA flow chart outlining the study selection process.

**Figure 2 cancers-16-01099-f002:**
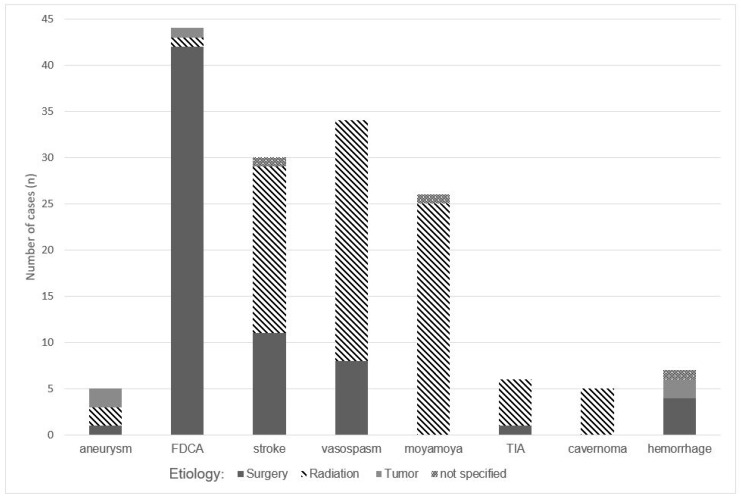
The number of published cases with neurovascular complications related to different etiologies. Abbreviations: FDCA: fusiform dilation of the carotid artery; TIA: transient ischemic attack.

**Figure 3 cancers-16-01099-f003:**
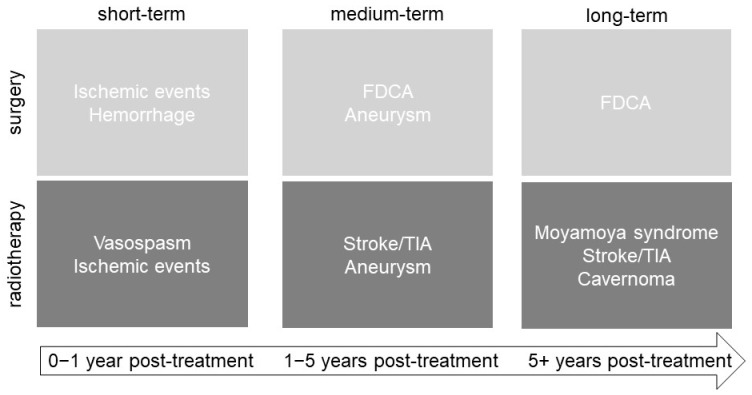
Short-, medium-, and long-term vascular complications reported due to the use of surgery and radiotherapy for treating craniopharyngioma.

**Table 1 cancers-16-01099-t001:** Eligibility criteria for record assessment process.

PCC Framework	Definition
Population	Patients with adamantinomatous or papillary CP with any age at diagnosis
Concept	Vascular damages, cerebro- and cardiovascular diseases, incidence, disease-specific mortality, risk factors, and outcome
Context	Case reports and observational studies published since 1990 in English or German

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
