# Peer review of "Vascular Morbidity and Mortality in Craniopharyngioma Patients—A Scoping Review"

_cancers, 2024, doi:10.3390/cancers16061099_

Round 1

Reviewer 1 Report

Comments and Suggestions for Authors

Beckaus et al. provides here a review of vascular complications in patients treated for craniopharyngioma.

This is a well-written and complete review covering adult as well as pediatric patients. 

I have 1 major suggestion and 2 minor suggestions:

1/ Vascular complications of treatment should also be classified according to the type of treatment received. For the surgery, the author should differentiate between upper (pterional / subfrontal) and transphenoidal (endoscopic / microscopic) approaches. Same observation for radiation therapy: stereotactic RT should be differentiated from other techniques. It appears that these factors are of importance for vascular insult.

2/ The simple summary is missing

3/ Please define MRA in the abstract as this abbreviation is not common

Reviewer 2 Report

Comments and Suggestions for Authors

The present scoping review about vascular and cardiac complications of CP patients is well written and conducted. On the other hand few problems are encountered as follow:

1) It is not always clear the relation between CP and some vascular complications/lesions (please discuss, argument, add details)

2) It is not clear if the authors consider all vascular lesions directly related to CP or if  they report simple/coincidental  association (please discuss, argument, add details).  

3) study limitations should be considered and discussed (please add)

Reviewer 3 Report

Comments and Suggestions for Authors

The scoping review presented ia a nice, well documented and ellaborated work resuming the neurovascular and cardiovascular risks of CP treatment related to surgery and radiotherapy.  It emphasizes the need to stay aware of these complications and include their search in active follow up programs including late MRAs  to diagnose neurovascular sequelae and aftercare prevention of cardiovasular risk factors. 

The quality of presentation is good both for the content and the form. The figures are nice and efficient. The tables are a bit dense but difficult to reduce. The references are well adjusted. 

I will just share some minor flaws and errors that I believe need revision:

- the simple summary is missing

- the second half of the abstract needs some rewriting to make sense. The sentences are written like a telegram

- line 26:  "i.a." should say "i.e."

- line 27: "(Cardio) vascular" should say "Cardiovascular"

- line 43:  "surgical resection via transsphenoidal and transcranial" shoyld say   " surgical resection via transsphenoidal or transcranial"

- line 155: "11% for cerebral infarction were found" should say "11% for cerebral infarction was found"

- In surgical related vascular sequelae, can you discuss if there is any risk  difference related to the approach (via craniotomy or transsphenoidal)?

- Until more  radioinduced cavernoma cases cases are followed it is maybe  not safe to conclude they are low risk. 

Comments on the Quality of English Language

Already told in the comments to authors:

- the second half of the abstract needs some rewriting to make sense. The sentences are written like a telegram

- line 26:  "i.a." should say "i.e."

- line 27: "(Cardio) vascular" should say "Cardiovascular"

- line 43:  "surgical resection via transsphenoidal and transcranial" shoyld say   " surgical resection via transsphenoidal or transcranial"

- line 155: "11% for cerebral infarction were found" should say "11% for cerebral infarction was found"

Reviewer 4 Report

Comments and Suggestions for Authors

 The work you do is necessary, but I have a few suggestions:
1.You should respect the requirements of the template and complete the Simple Summary paragraph. It has already been stated in the documentation that it will not be received without a Simple Summary.
2.You should modify the format of the image to make it more suitable for the file. The image currently appears in an awkward format, so you can consider rotating or adjusting it. Make sure that the image renders as expected in the file and that it clearly presents what you want.
3.As a review, references are important. While the current number of references is sufficient, you should consider the age of the references. In order to keep the information up-to-date and accurate, it is recommended that you cite more documents from the last three years rather than ancient ones. Remember, evaluate references for their authority, reliability, and relevance.

Comments on the Quality of English Language

 Extensive editing of English language required.

Round 2

Reviewer 1 Report

Comments and Suggestions for Authors

The authors responded to my main remarks

Reviewer 4 Report

Comments and Suggestions for Authors

The work has met the criteria for publication.

Comments on the Quality of English Language

MInor editing of English language required.